## COMMENT

# Counteracting deliberate ignorance of academic bullying and harassment

Konstantin Offer [1,2,3✉], Zoe Rahwan [1] & Ralph Hertwig [1]

Bullying and harassment are pervasive in academia, with many cases going unreported. One possible factor may be deliberate ignorance among perpetrators and bystanders. A number of interventions counteracting deliberate ignorance could contribute to thriving research environments.

Bullying and harassment remain significant problems in academia. Bullying—defined here as unwanted offensive and malicious behavior that undermines, patronizes, intimidates, or demeans the recipient or target[1]—may be direct (e.g., physical or verbal abuse) or indirect (e.g., setting unreasonable deadlines or withholding crucial information). Harassment is unwanted conduct related to a relevant protected characteristic (e.g., age, disability, or gender) that results in a hostile environment[2]. We focus on bullying and harassment within the educational and work context of academia, where high-profile cases of scientists making the headlines appear to be just the tip of the iceberg[3]: According to a 2019 synthesis of 70 empirical studies from 20 countries, on average, 25% of faculty self-identify as being bullied and 40–50% report having witnessed bullying within the past year[4]. Women, junior researchers, and members of minority groups are more likely to be bullied and harassed[1]. Moreover, many targets suffer persistent abuse (up to half for 3 years or more; 10–20% for 5 years or more)[4]. Yet only a minority of bullying and harassment cases are officially reported[5], with many targets hesitating to report mistreatment due to fear of retaliation or the belief that their concerns will go unheard[6].

**Deliberate ignorance**—defined as the conscious choice not to seek or use information—is known to serve important psychological and social functions, such as regulating emotions or avoiding liability[7]. Building on our research on deliberate ignorance, we make the conceptual argument that bystanders and perpetrators have distinct sets of psychological motives to remain ignorant of academic bullying and harassment (Fig. 1). We define perpetrators as individuals who have been established to have bullied or harassed others, most commonly in positions senior to targets[8]. Bystanders witness bullying or harassment without taking part in it. While many, though not all[9], institutions have introduced policies on how to respond to academic bullying and harassment, guidelines are not always properly enforced[10]. Furthermore, robust evidence on the efficacy of interventions is often lacking. We propose four institutional responses to better understand and counteract deliberate ignorance of academic bullying and harassment.

We acknowledge that targets of bullying and harassment may also engage in deliberate ignorance (e.g., downplaying the bullying as a coping mechanism to protect their mental health). Moreover, we recognize that academic bullying and harassment typically occur in public and institutional settings, meaning that they have dimensions beyond the individuals involved. We briefly touch upon institutional deliberate ignorance. Our focus here, however, is on understanding and addressing deliberate ignorance among perpetrators and bystanders.

---

[1] Center for Adaptive Rationality, Max Planck Institute for Human Development, Berlin, Germany. [2] Max Planck School of Cognition, Leipzig, Germany. [3] Department of Psychology, Humboldt-Universität zu Berlin, Berlin, Germany. ✉email: offer@mpib-berlin.mpg.de

  

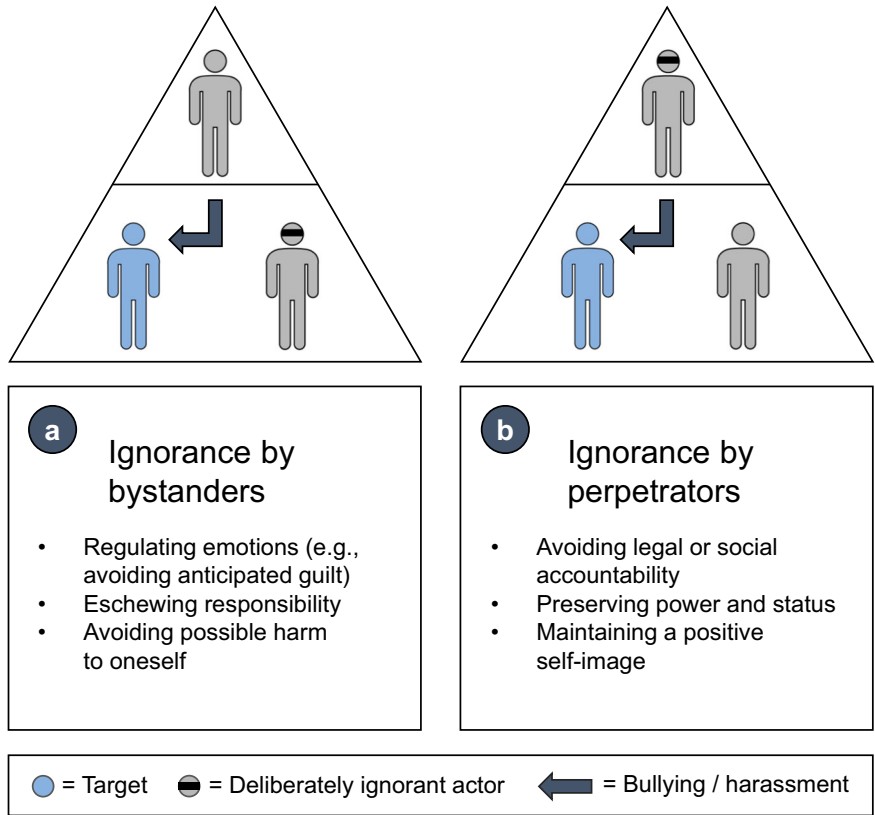

**Fig. 1 Deliberate ignorance of academic bullying and harassment.** Psychological motives for deliberate ignorance of academic bullying and harassment in (**a**) bystanders and (**b**) perpetrators. Although perpetrators are depicted here as having a higher status than targets and bystanders (downward bullying), other hierarchical configurations are possible. Less typically, perpetrators may have a similar or lower status than targets, and bystanders may have a similar or higher status than perpetrators. Thus subordinates can also be perpetrators in cases of upward bullying[8]. These hierarchical dynamics are expected to differentially affect the psychological motives for deliberate ignorance.

### Understanding ignorance

The bystander effect has been demonstrated in many studies[11]: The mere presence of bystanders in critical situations can reduce an individual's probability of helping. Classic explanations are twofold. First, the more people are present, the lower the experienced sense of personal responsibility. Responsibility diffuses. Second, almost all group members can privately reject a norm to help and, at the same time, believe that almost everyone else accepts it. Ignorance can be pluralistic. Recent research suggests that bystander ignorance may also be deliberate, with people having various psychological motives for turning a blind eye to misconduct. For example, consciously choosing not to seek information—one form of deliberate ignorance—can be a way of regulating one's emotions and deflecting responsibility[12]. Deliberate ignorance can help to avoid distress and the anticipated guilt for not getting involved. Consciously choosing not to act on relevant information—a second form of deliberate ignorance—may be used as a strategic device to eschew responsibility and to avoid possible harm to oneself (see Fig. 1).

Psychological motives for deliberate ignorance can depend on the bystander's status relative to the perpetrator. Strategic motives may be more pronounced in relationships with power asymmetries. For example, junior scientists may anticipate being unfavorably treated by a higher ranked perpetrator and remain deliberately ignorant to protect themselves. Emotion regulation may be a more significant motive when bystanders and perpetrators share a similar rank (e.g., a peer-to-peer relationship between two tenured professors). Witnessing a peer's unethical behavior can be distressing, and deliberate ignorance can help bystanders to regulate their fear of confrontation with a peer, their guilt for not helping a target, or both.

Perpetrators may choose to ignore the distressing and even traumatizing effects of their behavior on targets in an attempt to escape social or legal accountability[12]. In turn, this can preserve their power and status in academic hierarchies and help them maintain a positive self-image (see Fig. 1). We review policies that address deliberate ignorance in both perpetrators and bystanders and propose corresponding interventions intended to contribute to more ethical environments for all participants in academia.

### Counteracting ignorance

**Institutionalize regular organizational screenings**. Although aggregate data on academic bullying and harassment are available, many institutions remain unaware—wittingly or unwittingly—of the prevalence, causes, and consequences of bullying and harassment within their own ranks. After being shaken by two high-profile bullying allegations in 2018[13], Germany's Max Planck Society faced uncertainty about the scale of bullying and harassment across its institutes. In 2019, the Society conducted a staff survey on bullying and received more than 9000 responses (38% of staff). Higher rates of bullying were reported by non-scientific (12%) than scientific staff (8%), by women (12%) than men (8%), and by older than younger employees (e.g., 13% for ages 45–59 vs. 7% for ages 15–29)[5]. Since the survey, follow-up screenings have been introduced to monitor progress and changes in work culture. Other institutions, such as the University of California[1], have also engaged in comprehensive, large-scale screenings to systematically understand bullying and harassment behavior. The regular and direct seeking of such information actively counteracts deliberate ignorance across institutional

ranks. Yet organizational screenings are only effective if institutions also on the knowledge generated to create sustained change.

**Establish independent and anonymous reporting channels**. Although many institutions have ombudspersons or conflict management services in place, targets often hesitate to report abuse due to concerns about potential retaliation[6]. Yet targets should not be the only party reporting unethical behavior. Institutions need to encourage all members of their community to report observed misconduct and unethical practices. The onboarding of new employees can be a good opportunity to address ignorance by bystanders and inform staff about the reporting channels available to them. Uptake of these channels could be further encouraged by informing employees about the prevalence of bullying and harassment in the organization, as well as the psychological motives for not wanting to know. In particular, bystanders might be less likely to ignore bullying and harassment if they are educated on the problem of bystander ignorance and made aware of action pathways available to them. Further, institutions can benefit from offering external and independent reporting channels to prevent conflicts of interest. Finally, given that institutions may themselves engage willfully in ignorance to protect their reputation or maintain funding, guidelines requiring reporting of established instances of misconduct to third parties (e.g., funding agencies) may be beneficial.

**Provide robust whistle-blower protection**. One important psychological motive for bystanders not approaching targets and inquiring about their wellbeing is to avoid possible harm to themselves. This motive may be particularly pronounced when a perpetrator is more senior. Career progression in academia can depend on a senior scientist's support, particularly in close-knit fields or disciplines. Whistle-blowers, therefore, need special protection. Beyond legal protections and anonymous reporting systems, a robust whistle-blower protection system includes anti-retaliation policies, optional relocations and fall-back supervision agreements. Further, protection from emotional and mental harm can be supported through the institutional provision of free, anonymous, and independent counseling services. Witnesses who feel protected and have confidence that due process will be followed may be more likely to report unethical practices. This requires a firm stance at the institutional level, with clear and robust consequences for perpetrators (e.g., official reprimands, withdrawal of funding, or even dismissal) being established and enforced.

**Foster psychological safety by boosting individual competences**. The information gathered on bullying and harassment can be used to develop more effective interventions to foster psychological safety (i.e., shared beliefs that concerns can be expressed without fear of consequences). In particular, *boosting* approaches—behavioral interventions that foster people's competences to make their own decisions—provide potential for lasting behavioral change[14]. For example, active bystander trainings can equip employees with the skills to effectively challenge unacceptable behaviors. As they learn about the psychological motives for not seeking information, they may become more likely to stand up to abuse. Perpetrators may be ignorant of the impact of their behavior on others; evidence-based training programs drawing on screening results and confirmed cases can be designed to combat this deliberate ignorance. Research group leaders can hone their abilities to lead teams in fair, respectful, and ethically sound ways by discussing past harassment cases and learning about self-control strategies. While such boosts may have

a direct short-term impact, we see their main potential in supporting lasting change by equipping leaders with the necessary skills to combat academic bullying and harassment and by aligning values across generations, fostering a collective understanding and agreement on ethical principles that allow all participants in the research environment to thrive.

## Conclusion

Academic bullying and harassment can have devastating and long-lasting effects on individuals and institutions—yet many cases go unreported. Perpetrators and bystanders have distinct psychological motives for turning a blind eye to misconduct, which can be shaped by their relative positions in hierarchies. We argue, based on conceptual grounds, that institutions need to overcome deliberate ignorance by systematically generating knowledge about the prevalence, causes, and consequences of bullying and harassment and using the insights gained to develop more effective and evidence-based interventions. Using this lens to understand bullying and harassment, academic institutions may be better prepared to manage and ideally prevent misconduct.

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

## Acknowledgements

We thank Susannah Goss for editing the manuscript. This work was supported by the Federal Ministry of Education and Research (BMBF) and the Max Planck Society. We acknowledge financial support from the Max Planck Institute for Human Development. The funders had no role in the preparation of the manuscript or the decision to publish.

## Author contributions

K.O. developed the research idea and wrote a first draft of the Comment, which was jointly revised by K.O., Z.R., and R.H.

## Funding

## Competing interests

The authors declare no competing interests.
