## [Peer Review File · Communications Psychology]

2nd Nov 23

Dear Mr Offer,

Your Comment titled "Understanding and Counteracting Deliberate Ignorance of Bullying and Harassment in Academia" has now been seen by 2 referees, whose comments appear below. In light of their advice, I am delighted to say that we are happy, in principle, to publish it in Communications Psychology under a Creative Commons 'CC BY' open access license.

We will not send your revised paper for further review if, in the editors' judgment, the referees' comments on the present version have been addressed. If the revised paper is in Communications Psychology format, in an accessible style, and of appropriate length, we shall accept it for publication immediately. To facilitate your response to the referees and meet our formatting requirements, I have attached an edited version of your manuscript, and ask you to attend to each comment in detail.

EDITORIAL REQUESTS:

* Please review the changes in the attached copy of your manuscript, which has been edited for style, and address the comments and queries I have added. If using Word, please use the 'track changes' feature to make the process of accepting your manuscript more efficient.

* Please check whether your manuscript contains third-party images, such as figures from the literature, stock photos, clip art or commercial satellite and map data. If any of the display items in your manuscript (figures, tables, boxes or movies) include images that are the same as, or are adaptations of, previously published images, please fill in the ``Third Party Rights Table``, and return to us when you submit your revised manuscript. This information will enable us to obtain the necessary rights to re-use such material. If we are unable to obtain the necessary rights to use or adapt any of the material that you wish to use, we will contact you to discuss alternative options.

* Communications Psychology uses a transparent peer review system. On author request, confidential information and data can be removed from the published reviewer reports and rebuttal letters prior to publication. If you are concerned about the release of confidential data, please let us know specifically what information you would like to have removed. Please note that we cannot incorporate redactions for any other reasons.

*If you have not done so already, please alert me to any related manuscripts from your group that are under consideration or in press at other journals, or are being written up for submission to other journals (see www.nature.com/authors/editorial_policies/duplicate.html for details).

FORMATTING GUIDELINES:

You will find a complete list of formatting requirements following this link:

<https://www.nature.com/documents/commsj-style-formatting-checklist-comment.pdf>

Please use the checklist to prepare your manuscript for final submission. In the following, I also

highlight some issues of particular importance.

**** Title**

Titles should be descriptive of the main message your manuscript conveys and should not exceed 90 characters (including spaces). Please note that punctuation is not allowed, nor are titles of the following format: "title: subtitle".

**** Preface**

The paper's preface (up to 40 words; without references) should serve both as a general introduction to the topic, and highlight your position or proposal. Because we hope that researchers across all fields of psychology will be interested in your work, the preface should be as accessible as possible.

**** Length**

The ideal length for Comment articles in Communications Psychology is 1,500 words. We have some flexibility, however, but please ensure that your text does not exceed its present length (1,807).

**** Main text**

Please provide three or four section headings in the main text. These should relate to the content of the article rather than being generic. Headings should be no longer than 30 characters (including spaces) and should not use punctuation.

**** Figures**

Please remove all figures from the main text and upload them individually, one figure per file. To ensure the swift processing of your paper please provide the highest quality, vector format, versions of your images (.ai, .eps, .psd) where available. Text and labelling should be in a separate layer to enable editing during the production process. If vector files are not available then please supply the figures in whichever format they were compiled in and not saved as flat .jpeg or .TIFF files. If your artwork contains any photographic images, please ensure these are at least 300 dpi.

* Figures should be simple and informative — multi-part figures are best avoided.

*** References**

The limit of 15 References is strict. I included in the Manuscript guidance on which References may be removed.

References appear as superscript Arabic numerals, in order of mention. The reference list mentions references in the numerical order in which they are mentioned in the main text. If a reference is cited more than once, the same number is used throughout the text and the reference receives a single entry in the reference list.

Only papers that have been published or accepted by a named publication should be in the reference list (preprints and citations of datasets are also permitted). Unpublished/Submitted research should not be included in the reference list; it should only be mentioned briefly and parenthetically in the main text. Note that no major arguments should rely on unpublished research.

Published conference abstracts and URLs for web sites should be cited parenthetically in the text, not in the reference list.

Footnotes are not used.

* Competing interests

Please include a "Competing interests" statement after the References. Note that we ask authors to declare both financial and non-financial competing interests. For more details, see <https://www.nature.com/authors/policies/competing.html>. If you have no financial or non-financial competing interests, please state so: "The authors declare no competing interests."

SUBMISSION INFORMATION:

* Your paper will be accompanied by a two-sentence editor's summary, of between 250-300 characters, when it is published on our homepage. We will usually use the Preface for that purpose.

In order to accept your paper, we require the following:

* A cover letter describing your response to our editorial requests.

* The final version of your text as a Word or TeX/LaTeX file, with any tables prepared using the Table menu in Word or the table environment in TeX/LaTeX and using the 'track changes' feature in Word.

* Production-quality versions of all figures, supplied as separate files. Photographic images should be 300 dpi in RGB format (.jpg, TIFF or native Photoshop format) and any labels/scale bars included in a separate layer from the image. Line art, graphs and schemes should be vector format (.ai, .eps, .pdf); Adobe Illustrator files are preferred and will minimize production time. Any chemical structures or schemes contained within figures should additionally be supplied as separate Chemdraw (.cdx) files.

At acceptance, the corresponding author will be required to complete an Open Access Licence to Publish on behalf of all authors, and declare that all required third-party permissions have been obtained.

Please note that your paper cannot be sent for typesetting to our production team until we have received this information; **therefore, please ensure that you have this ready when submitting the final version of your manuscript.**

ORCID

Communications Psychology is committed to improving transparency in authorship. As part of our efforts in this direction, we are now requesting that all authors identified as 'corresponding author' create and link their Open Researcher and Contributor Identifier (ORCID) with their account on the Manuscript Tracking System (MTS) prior to acceptance. ORCID helps the scientific community

achieve unambiguous attribution of all scholarly contributions. For more information please visit <http://www.springernature.com/orcid>

For all corresponding authors listed on the manuscript, please follow the instructions in the link below to link your ORCID to your account on our MTS before submitting the final version of the manuscript. If you do not yet have an ORCID you will be able to create one in minutes.

IMPORTANT: All authors identified as 'corresponding author' on the manuscript must follow these instructions. Non-corresponding authors do not have to link their ORCIDs but are encouraged to do so. Please note that it will not be possible to add/modify ORCIDs at proof. Thus, if they wish to have their ORCID added to the paper they must also follow the above procedure prior to acceptance.

To support ORCID's aims, we only allow a single ORCID identifier to be attached to one account. If you have any issues attaching an ORCID identifier to your MTS account, please contact the Platform Support Helpdesk.

[link redacted]

We hope to hear from you within two weeks; please let us know if the process may take longer.

Best wishes,

Marike

Marike Schiffer, PhD
Chief Editor
Communications Psychology

REVIEWERS' EXPERTISE:

Reviewer #1: bullying and harassment research, advocacy

Reviewer #2: bullying and harassment research

REVIEWERS' COMMENTS:

Reviewer #1 (Remarks to the Author):

I enjoyed reading this comment regarding the the current deliberate ignorance of te incidences of academic bullying and harassment. I publish this informative paper after minor revision noted below:

1- The solutions provided are mostly focused on institutional actions; institutions, however, have limited tendency to protect targets and face perpetrators due to several reasons (e.g., their reputation and the funding that a bully may bring to the universities). I encourage the authors to be more focused on integrated functioning of various stakeholders (e.g., funding agencies and policy makers) in response to the deliberate ignorance of bullying incidences.

2- The readers may also benefit from a short discussion that the deliberate ignorance can happen in both upward and downward bullying and harassment.

3- The role of the members of the internal investigation committee is another ignored factor in the deliberate ignorance of academic bullying and harassment (this ref might be relevant: <https://www.scientificamerican.com/article/shoddy-harassment-investigations-are-a-stain-to-academia/>)

Reviewer #2 (Remarks to the Author):

The authors posit a unique perspective to explain why bullying and harassment occur in academia—that of deliberate ignorance. Bullying and harassment have garnered attention in the literature, particularly as it relates to the general non-academic workforce and for trainees. However, the authors provide not only one idea as to why these mistreatments perpetuate, but also what can be done about it at an institutional and individual level.

I do have suggestions for the authors to improve this Comment.

1. Standfirst: The last sentence says “support all researchers to thrive.” I encourage the authors to be more inclusive and think about academia beyond researchers by including faculty, clinicians, and trainees. All these individuals play a crucial role in academics and can also be greatly impacted by bullying and/or harassment.

2. Introduction: There are more comprehensive definitions of bullying than what is presented. One recent definition by Averbuch et al states “bullying is the conscious action of offenders abusing positions of authority and intentionally targeting individuals through persistent negative behaviors to impede education or career growth” (reference: Averbuch T, Eliya Y, Van Spall HGC. Systematic review of academic bullying in medical settings: dynamics and consequences. *BMJ Open*. 2021 Jul 12;11(7):e043256.) The intentionality of impeding career growth is crucial to the definition and is missing from the authors’ definition. In addition, the authors’ definition does not include that there is often a power differential between the perpetrator and target. The authors do allude to this power differential later, but including it in the definition influences future interventions.

3. Introduction: The authors list references that while not outdated, are not as recent as others (especially references 5-7). Alternative references include: Iyer MS, Way DP, MacDowell DJ, Overholser BM, Spector ND, Jagsi R. Bullying in Academic Medicine: Experiences of Women Physician Leaders. *Acad Med*. 2023 Feb 1;98(2):255-263; Iyer MS, Way DP, MacDowell DJ, Overholser B, Spector ND, Jagsi R. Why Gender-Based Bullying Is Normalized in Academic Medicine: Experiences and Perspectives of Women Physician Leaders. *J Womens Health (Larchmt)*. 2023 Mar;32(3):347-355.

4. Introduction: The authors write, “Although many institutions have now introduced policies on how to respond to bullying and harassment in academia, guidelines are not always properly enforced.” What is even more important is that many medical schools (as an example of institutions) do not even have antibullying policies, which could/should be included here (reference: Iyer MS, Choi Y, Hobgood C. Presence and Comprehensiveness of Antibullying Policies for Faculty at US Medical Schools. JAMA Netw Open. 2022 Aug 1;5(8):e2228673)

5. Deliberate Ignorance. This principle may definitely apply to both targets and also bystanders when they consciously overlook bullying/harassment or when they do not report such behaviors. However, bullies themselves may be deliberate in their actions but ignorant in knowing that they are bullies. I am not entirely sure that deliberate ignorance is the principle that the bullies use to perpetuate their power. For example, on page 5, the authors write “By choosing to ignore the distressing and even traumatizing effects of their behaviour on targets, perpetrators may attempt to escape social or legal accountability. The bullies are often ignorant of the impact of their behaviors but deliberate in their actions. One reference that highlights this as well as the fact that there is a cycle to bullying (bullying starts at a young age or the bullied becomes the bully is Harvey, M. G., Heames, J. T., Richey, R. G., & Leonard, N. (2006). Bullying: From the Playground to the Boardroom. Journal of Leadership & Organizational Studies, 12(4), 1-11.

6. Institutionalize Regular Organizational Screenings: Suggest adding to the last sentence of this paragraph that such screenings are only effective if leadership uses the results to implement change.

7. The authors describe the following four interventions: Institutionalize regular organizational screenings, Establish Independent and Anonymous Reporting Channels, Provide Robust Whistleblower Protection, and Fostering Individual Psychological safety. However, the authors are missing an entire section on zero tolerance policies (which includes step wise approach to have conversations with the bully, discussions about next steps in compensations etc, and consequences such as termination) that must be in place in first so that the interventions mentioned by the authors are successful.

Kudos to the authors for NOT suggesting that an antibullying or antiharassment intervention includes remediation of the target. Often times, targets are in near impossible situations to advocate for themselves. Finally, I do believe that this Comment will be valuable to a wide audience and even more improved, if the changes above are incorporated.